Relating form to function in the hummingbird feeding apparatus

Rico-Guevara Alejandro a.rico@berkeley.edu
Department of Integrative Biology, University of California , Berkeley , CA , United States of America
Abdala Virginia
Electronic publication date: 2017 Jun 8
Publication date: 2017
Volume: 5
Electronic Location ID: e3449
Received 2017 Apr 14; Accepted 2017 May 19
Copyright: ©2017 Rico-Guevara
Copyright year: 2017
Copyright holder: Rico-Guevara
License: This is an open access article distributed under the terms of the Creative Commons Attribution License, which permits unrestricted use, distribution, reproduction and adaptation in any medium and for any purpose provided that it is properly attributed. For attribution, the original author(s), title, publication source (PeerJ) and either DOI or URL of the article must be cited.
License URL: https://creativecommons.org/licenses/by/4.0/

Keywords: Anatomy, Bill, Computed tomography, Electron microscopy, Lingual apparatus, Nectar feeding, Tongue

Funding: American Ornithologists’ Union Ecology and Evolutionary Biology Department at the University of Connecticut Miller Institute This study was funded by the American Ornithologists’ Union, the Ecology and Evolutionary Biology Department at the University of Connecticut, and the Miller Institute. The funders had no role in study design, data collection and analysis, decision to publish, or preparation of the manuscript.

==============================
A complete understanding of the feeding structures is fundamental in order to study how animals survive. Some birds use long and protrusible tongues as the main tool to collect their central caloric source (e.g., woodpeckers and nectarivores). Hummingbirds are the oldest and most diverse clade of nectarivorous vertebrates, being a perfect subject to study tongue specializations. Their tongue functions to intraorally transport arthropods through their long bills and enables them to exploit the nectarivorous niche by collecting small amounts of liquid, therefore it is of vital importance to study its anatomy and structure at various scales. I focused on the portions of the hummingbird tongue that have been shown to be key for understanding their feeding mechanisms. I used histology, transmission and scanning electron microscopy, microCT, and ex-vivo experiments in order to advance the comprehension of the morphology and functioning of the hummingbird feeding apparatus. I found that hummingbird tongues are composed mainly of thin cornified epithelium, lack papillae, and completely fill the internal cast of the rostral oropharyngeal cavity. Understanding this puzzle-piece match between bill and tongue will be essential for the study of intraoral transport of nectar. Likewise, I found that the structural composition and tissue architecture of the tongue groove walls provide the rostral portion of the tongue with elastic properties that are central to the study of tongue-nectar interactions during the feeding process. Detailed studies on hummingbirds set the basis for comparisons with other nectar-feeding birds and contribute to comprehend the natural solutions to collecting liquids in the most efficient way possible.

Introduction

A central challenge of biological studies is to describe the links among the structures (e.g., organismal morphology), underlying mechanisms (e.g., biomechanics), and emergent phenomena (e.g., performance, ecological and evolutionary patterns) in live organisms. Birds are an ideal subject to tackle this challenge since they have evolved the most morphologically diverse array of feeding structures among tetrapods (Rubega, 2000). A thorough understanding of the form and function of the feeding structures is vital to grasp the functional constraints that steer the evolution of resource exploitation in animals. In birds, it has been recognized that bill shape is tightly correlated to diet (cf. Rubega, 2000), however, this idea has been challenged in raptorial birds by the correlation between skull and beak structure implying developmental constraints (Bright et al., 2016). It has been highlighted recently that (1) phylogeny and allometry are determinants in the variation of bill shape, with high diversification rates at the dawn of modern birds followed by a slowed down diversification phase of morpho-space packing (Cooney et al., 2017), and that (2) to fully understand the evolution of the feeding apparatus a reappraisal of the linguo-laryngeal system in the context of the skull-beak coupling is warranted (Homberger, 2017). If bill shape provides information about generally which type of food is consumed (e.g., seeds vs. meat); as a complement, I hypothesize that lingual apparatus morphology could provide further information about how the food is consumed. Examples can be found in the extreme reduction of the tongue of cormorants (Jackowiak, Andrzejewski & Godynicki, 2006), the gigantic papillae of penguins (Kobayashi et al., 1998), and the numerous flexible projections of flamingo tongues (Zweers et al., 1995). Avian tongues present adaptations as extensive and varied as those of bird bills (Farner, 1960). Unveiling the details of the morphology and coupling of the components of the feeding apparatus advances the understanding of its function and evolution.

Birds control the movement of their tongues with muscles attached to the hyobranchial apparatus (set of supporting bones); these ‘intrinsic hyolingual muscles’ (Homberger & Meyers, 1989; Tomlinson, 2000; but see Schwenk, 2001) have their most rostral attachments on a paired bone called the paraglossum (cf., Weymouth, Lasiewski & Berger, 1964; or Os entoglossum, Newton et al., 1896). Some birds, such as woodpeckers (Shufeldt, 1900; Villard & Cuisin, 2004) and nectar-feeding birds (Stiles, 1981; Paton & Collins, 1989), have to protrude their tongues to procure their food. Interestingly, woodpeckers have the ability to actively control their tongue tips (cf. Bock, 1999), a capacity that is lacking in hummingbirds (Zusi, 2013). The reason for this dissimilarity relies on the differential elongation of the tongue components; in woodpeckers, the portion of the tongue supported by the paraglossum is not elongated whereas in hummingbirds this portion is greatly lengthened. In most birds, only the rostral third of the tongue is entirely free of musculature (review in Erdoğan & Iwasaki, 2014), but in hummingbirds between half (Scharnke, 1931; Weymouth, Lasiewski & Berger, 1964) to three fourths (Rico-Guevara, 2014) of the tongue lacks muscles, bone, and/or cartilage support. Only a pair of cornified rods at the lingual tip (cf. Weymouth, Lasiewski & Berger, 1964) provides rigidity to the rostral membranous tube-like grooves in hummingbird tongues (Fig. 1 in Rico-Guevara & Rubega, 2011). It is puzzling that this highly specialized food collection tool lacks active control, and it is important to understand how tissue organization and properties alone govern the tongue functioning in nectar collection.

In birds, the diversity in feeding apparatus came with niche specialization; as one of the prime examples, primitive insectivorous hummingbirds entered the nectar-feeding niche and became one of the most specialized nectarivorous vertebrates (Stiles, 1981; Fleming & Muchhala, 2008; Baldwin et al., 2014). Early hummingbirds rapidly acquired a novel bill shape (diverging from the wide and short beak typical of Strisores) that fostered faster morphological diversification than the one experienced by the rest of the birds (Cooney et al., 2017) via coevolution with flowers (Stiles, 1981; Weinstein & Graham, 2017) and the development of a wide array of foraging strategies (Feinsinger & Colwell, 1978) linking exploitative and interference competition to extreme bill structural configurations (e.g., Rico-Guevara, 2014; Remsen Jr, Stiles & Mcguire, 2015). Hummingbirds still catch insects as their main source of protein, exhibiting a variety of hunting tactics (e.g., Stiles, 1995; Rico-Guevara, 2008) and using their tongues to drag prey they catch near their bill tips to where it can be swallowed (e.g., Yanega, 2007). Therefore, they use their tongue protrusion abilities for both arthropod intraoral transport and nectar collection (e.g., Rico-Guevara, 2014). Although hummingbird tongues have been studied for about two centuries (Martin, 1833; Darwin, 1841; Lucas, 1891; Scharnke, 1931; Weymouth, Lasiewski & Berger, 1964; Hainsworth, 1973), many aspects of their morphology and function still remain to be understood. The tongues of hummingbirds are forked at their tips (Martin, 1833; Darwin, 1841; Scharnke, 1931; Hainsworth, 1973), ending in two tube-like grooves with fringed edges (Lucas, 1891). These grooves are exclusively rostral structures and the interior of the tongue base is not hollow (Scharnke, 1931; Weymouth, Lasiewski & Berger, 1964). There is only one study focusing on the morphology of the entire length of the tongue grooves (Hainsworth, 1973), which unfortunately is lacking histological details. The most rostral cross section micrograph near the base of the tongue grooves (Weymouth, Lasiewski & Berger, 1964) shows at least two distinct layers of tissue composing the dorsal and ventral surfaces of the grooves, which are not further described. Studies on nectar feeding in living birds suggest that the functional traits enabling hummingbirds to extract liquid are related to the structural configuration of the tongue tip (Rico-Guevara & Rubega, 2011; Rico-Guevara, Fan & Rubega, 2015), rather than to active movements of their parts through muscle action. A deeper study of the entire length of hummingbird tongues is essential to understand the underlying architectural properties enabling the observed nectar extraction mechanisms. Because previous studies (e.g., Weymouth, Lasiewski & Berger, 1964; Zusi, 2013) have described in detail the hyobranchial apparatus, the structure of the root, and the body of the tongue (up to the bifurcation point) in hummingbirds, the present study presents only descriptions of the structures of the rostral portion of the tongue grooves, and in addition, a description of the coupling between the bill and tongue. Understanding the morphology of the rostral portion of the grooves and the bill-tongue fit is crucial to understand the nectar-feeding mechanics in hummingbirds (e.g., Rico-Guevara, 2014). Furthermore, because the proposed mechanism of nectar collection involves passive transformations of the tongue modulated by the interaction with the bill tips (Rico-Guevara & Rubega, 2011), it is not enough to understand the morphology of each interacting part, but also it is necessary to study their functioning. Since the tongue transformations are purported as passive, in theory they could be replicated under laboratory conditions thus validating or rejecting previously proposed biomechanical hypotheses (e.g., Rico-Guevara, Fan & Rubega, 2015).

The aims of this paper are (1) to provide a description of the coupling of the components of the feeding apparatus in hummingbirds –namely the bill-tongue three-dimensional fit, (2) to describe the tissue architecture and surfaces of the tongue tip, (3) to characterize and contextualize the gross and detailed morphology of the hummingbird feeding apparatus both in a comparative (among birds) and ecologically relevant (biomechanics) framework, and (4) to perform experiments to reveal the extent to which the feeding structures can passively transform to contribute in the nectar collection process (i.e., post-mortem experiments). I used histology, transmission and scanning electron microscopy, and high-resolution X-ray computed tomography (microCT) to describe larger anatomical features and the three-dimensional arrangement of the tongue inside the bill (Fig. 1, Video S1). There have been few studies, like the one presented here, that merged microCT, light, and electron microscopy in order to examine morphological features by linking them across disparate spatial scales (Handschuh et al., 2013; Jung et al., 2016).

Figure 1 Depiction of the techniques used to study the hummingbird feeding apparatus.

(A) Photograph of a hovering Anna’s Hummingbird (Calypte anna, courtesy of Robert McQuade) with an overimposed microCT 3D reconstruction of its bill. (B) MicroCT scan coronal cutaway section portraying both the bill and tongue. (C) MicroCT scan reconstruction depicting a section of the tongue. (D) Light microscopy photograph portraying a section of the tongue with the supporting rod at the top. (E) Electron microscopy photograph depicting a section of the tongue wall tissue to show its architecture.

Materials & Methods

I dissected five Ruby-throated Hummingbirds (Archilochus colubris Linnaeus, 1758), one Rufous Hummingbird (Selasphorus rufus Gmelin, 1788), one Anna’s Hummingbird (Calypte anna Lesson, 1829), one Short-tailed Woodstar (Myrmia micrura Gould, 1854), one White-necked Jacobin (Florisuga mellivora Linnaeus, 1758), and one White-tipped Sicklebill (Eutoxeres aquila Bourcier, 1847), for a total of ten specimens from six hummingbird species encompassing different clades in the hummingbird phylogeny. Four of the studied species (genera Archilochus, Selasphorus, Calypte, and Myrmia) belong to the most specious (high rate of diversification) clade named Bees, and the other two belong to more basal splits and least specious clades; Florisuga in the Topazes, and Eutoxeres in the Hermits (McGuire et al., 2014). I do not present phylogenetic comparative methods because all imaging techniques were not used for all the species (see below), the results presented here are descriptive, and it is not the aim of this paper (see ‘Introduction’). The inferences drawn from each method apply specifically to the species specified in each case, and unless stated in the text I do not present data on interspecific variation. All of the specimens were received as donations (e.g., dying birds that could not be rehabilitated) to the ornithological collections at the Department of Ecology and Evolutionary Biology of the University of Connecticut and at the Instituto de Ciencias Naturales of the National University of Colombia, between January 2012 and August 2013 and coming from several locations in the US, Colombia, and Ecuador. I only dissected (and processed as described below) recently deceased specimens ensuring that the tissues were fresh at the moment of each sample preparation. Once the investigation was concluded, the specimens were deposited in the freezers of the research laboratories at both universities (given the restrictions of the specimen preparations, see below) and are waiting for accession numbers and the development of specific collections for this kind of subjects. Electron microscopy specimens were deposited at the Bioscience Electron Microscopy Laboratory at the University of Connecticut. All activities in this study were reviewed and authorized by the Institutional Animal Care and Use Committee at the University of Connecticut; Institutional Animal Care and Use Committee Exemption Number E09-010. The anatomical nomenclature follows Nomina Anatomica Avium (Baumel et al., 1993, also see Homberger, 2017).

High-resolution X-ray computed tomography (microCT)

I dissected three salvaged specimens, a Ruby-throated Hummingbird, an Anna’s Hummingbird, and a Short-tailed Woodstar to scan their heads. Such dissections consisted of separating the head from the rest of the body, which allowed a more expedited and low-cost staining procedure (see below) and a better positioning of the specimens for the scanning process (closer to the X-ray source to achieve higher resolution). To obtain detailed morphological data at the micrometric scale and visualize the tongue soft tissues, I employed a staining protocol with osmium tetroxide (OsO4, cf. Metscher, 2009) with the difference that I did not embed my samples in resin, but instead placed them in small vials that could be positioned as close to the X-ray emitter as required for the desired resolution. I opted for osmium instead of iodine (e.g., Lautenschlager, Bright & Rayfield, 2014) because, although they both seem to bind to lipids (Bozzola & Russell, 1999; Gignac & Kley, 2014), osmium stabilizes tissue proteins, which then do not coagulate during dehydration with alcohol (Hayat, 2000). The heads were kept in 10% neutral buffered formalin and fixed with a solution containing 2.5% (wt/vol) glutaraldehyde and 2% (wt/vol) formaldehyde in 0.1 M sodium cacodylate trihydrate buffer (pH 7.4 adjusted with NaOH) for 8 h at 4 °C. After two washes in distilled water, the heads were fixed/stained with 2% (wt/vol) OsO4 in 0.1 M cacodylate buffer water for 4 h at 4 °C. Samples were washed three times in distilled water (20 min apart at 4 °C) and then dehydrated in a graded series of ethanol solutions. The specimens were stored in 100% ethanol at 4 °C and scanned at The University of Texas High-Resolution X-ray Computed Tomography Facility. Scans were performed at 70 kV and 10 W, with Xradia 0.5 and 4X objectives, and 1 mm SiO2, or no filter. Specimens were scanned in three parts, scans were stitched using Xradia plugins, and voxel size was between 15.5 and 5.2 µm. I obtained 16bit TIFF images that were reconstructed by Xradia Reconstructor, and the total number of slices per specimen was between 2,223 and 2,854, with scan times between 4 and 7 h. Using the data from the microCT scans I digitally decoupled the feeding apparatus components (segmenting in Avizo©) and constructed three-dimensional models to study the bill and tongue match.

Histological preparations

I dissected two Ruby-throated Hummingbirds to extract their tongues, which were cut into ∼3-mm long sections and fixed with 1.5% (wt/vol) glutaraldehyde—1.5% (wt/vol) paraformaldehyde in standard buffer (0.1 M HEPES, 80 mM NaCl, 3 mM MgCl2, pH 7.4 adjusted with NaOH) for a total of 9 h at 4 °C with one change into fresh fixative after one hour. The sections were then fixed in a solution of 1% OsO4—0.8% potassium ferricyanide—0.1 M sodium cacodylate—0.375 M NaCl for 2 h at 4 °C and then washed in distilled water. The sections were dehydrated in a graded series of ethanol solutions, and embedded in epoxy resin (a mixture of Embed812, Araldite 502 and DDSA, blocks polymerized at 60 °C for 48 h). I obtained semi-thin cross sections (1 µm) that were stained with methylene blue/azure II (1:1) followed by counterstaining with fuchsine for light microscopy. Photomicrographs were captured using a JVC High Resolution CCTV digital camera on an Olympus BX51 compound microscope at different magnifications (up to 1,000x). I used Auto-Montage software (Syncroscopy Inc.) to compile images of multiple optical planes, thereby obtaining pseudo-planar fields of view with improved visualization of the tissue structures.

Transmission electron microscopy (TEM)

I used one Ruby-throated Hummingbird for TEM. Using some of the fixed and embedded sections (epoxy resin processed in a Microwave Tissue Processor, Pelco Biowave Pro) of the tongue from the histological preparations, I obtained thin (80-nm) cross sections using a diamond knife on a Leica Ultracut UCT Ultramicrotome. The sections were put on Formvar support films for TEM and stained with either 2% uranyl acetate (UA) and lead citrate (LC, Reynolds, 1963), UA LC and RuO4 vapors, or RuO4 vapors only (Xue, Trent & Osseo-Asare, 1989). These sections were then imaged at the Bioscience Electron Microscopy Laboratory at the University of Connecticut, with a FEI Tecnai G2 Spirit BioTWIN transmission electron microscope at an accelerating voltage of 80 kV and at direct magnifications up to 120,000×.

Scanning electron microscopy (SEM)

I dissected one Ruby-throated Hummingbird and one Rufous Hummingbird to extract their tongues. The tongues were flattened with microslides and fixed with a solution containing 2.5% (wt/vol) glutaraldehyde and 2% (wt/vol) paraformaldehyde in 0.1 M sodium cacodylate trihydrate buffer (pH 7.4 adjusted with NaOH) for 8 h at 4 °C. After six washes (30 min apart) with the 0.1 M cacodylate buffer, the tongues were fixed/stained with 2% (wt/vol) OsO4 (2.5 ml) in 0.1 M cacodylate buffer (1.7 ml) + distilled water (0.8 ml) for 8 h at 4 °C. The tongues were cleaned by washing them three times in the cacodylate buffer and then dehydrated in a graded series of ethanol solutions. For all of these washes I used jets of fluid (using droppers immersed in the liquids) to ensure that the tongues were free of debris (and remaining nectar) in both dorsal and ventral surfaces; I did not scrape the tongue surfaces in order to keep them intact for posterior visualization. The first tongue was dried with a critical point dryer (Polaron E3000) for 2 h. Unfortunately, critical point drying (CPD) caused the edges of the tongue in the rostral region (where it forms the grooves) to spiral inward while drying, and only a small proportion of the dorsal surface of the tongue was visible after CPD. For the second tongue, I opted to use nylon mesh biopsy capsules and tissue cassettes to keep the tissue from spiraling inward. I inserted the tissue between layers of filter paper (chemically stable and allows adequate fluid exchange) to prevent mechanical damage from the mesh. By employing SEM, I could visualize and photograph the regions of interest, including equal access to both dorsal and ventral surfaces.

After CPD, I sputter coated (Polaron E5100) the tongues with gold and palladium, and attached them to aluminum SEM stubs using double-sided carbon tape, coated the caudal ends of the tongues with silver paint, and connected them to the aluminum stubs in order to reduce charging effects. I imaged the tongues at the Bioscience Electron Microscopy Laboratory at the University of Connecticut, with a Zeiss DSM982 field emission scanning electron microscope operated at an accelerating voltage of 2 kV and at direct magnifications up to 50,000×.

Ex-vivo experiments

I dissected one Ruby-throated Hummingbird to examine tongue-nectar interactions post-mortem. Under an Olympus SZX-12 dissecting microscope, I attached a Micro-Manipulator Model FX-117 (Electron Microscopy Sciences©) via surgical micro clamps to the epibranchial bones of the hyobranchial apparatus (Fig. S1). I held the skull in place with articulating arms coupled to a soft “helmet” made out of a polyvinyl chloride sheet and an Irwin© Quick-Grip Mini Handi-Clamp with swiveling clamping pads provided with longitudinal and transversal furrows that matched the hummingbird’s bill basal diameter without compressing it. At the tip of the bill I positioned a Mitutoyo© Digimatic Digital Caliper connected to a laptop to compare the compression of the tongue by the bill tip in this artificial setting and match it with previous estimates in living hummingbirds (Rico-Guevara, Fan & Rubega, 2015). The end result was the ability to precisely control tongue flattening and protrusion (Video S2). I attached a second Micro-Manipulator to a reservoir filled with artificial nectar (18.6% sucrose concentration) in order to control the bill tip to nectar surface distance without moving the fixed head. Lastly, I filmed the tongue-nectar interactions by coupling a high-speed camera (TroubleShooter HR), running up to 1,260 frames/s (1,280 × 512 pixels), to the dissecting microscope.

Activities were reviewed and authorized by the Institutional Animal Care and Use Committee at the University of Connecticut; Exemption Number E13-001.

Figure 2 Selected feeding apparatus cross sections (1–10) from a microCT scan of an Anna’s Hummingbird (Calypte anna).

Black structure in the middle of the figure is a lateral view of the bill from the reconstructed scan, and the dashed orange lines crossing it correspond to the numbered cross sections. Upper and lower bills (rhinotheca and gnathotheca are the keratinous sheaths of the maxillary and mandibular bones respectively) on each section appear separated but in a living hummingbird they can be fully coupled when the bill is shut, leaving virtually no space outside the tongue grooves in the rostral region. Relevant structures for understanding the feeding apparatus functioning are labeled (see text).

Results

High-resolution X-ray computed tomography (microCT)

I present the first complete cross-section series of a hummingbird feeding apparatus. I started with the most caudal section at the nasal operculum (Fig. 2, cross section [XS] 1) where the tongue is dorso-ventrally flattened, and the tongue body (corpus linguae) has started to divide medially due to an ingrowth (sulcus linguae) of the dorsal and ventral epithelia (Fig. 2, XS 1; cf. XS 11 in Weymouth, Lasiewski & Berger, 1964). The tongue body in hummingbirds encompasses the tongue from a distinct base, at the joint between the basihyale and the paraglossum, to the rostral grooves. I do not present a description of the structure of the lingual body in this paper given that this has been detailed previously (Weymouth, Lasiewski & Berger, 1964). At XS 2 there is a dark layer of cornified tissue almost completely surrounding the lingual body. Such layers become thicker at the ingrowth region and eventually connect, when moving rostrally through cross sections (Fig. 2, XS 2–5), effectively dividing the tongue body (cf. XS 13 in Weymouth, Lasiewski & Berger, 1964) and giving rise to a bifid tongue. At XS 3 the semi-cylindrical configuration characteristic of the tongue grooves is already conspicuous (cf. XS 14 in Weymouth, Lasiewski & Berger, 1964).

At XS 4 it is apparent that the tissue inside the lingual body chambers is thinner, leaving an empty space dorso-laterally (cf. XS 15–17 in Weymouth, Lasiewski & Berger, 1964). At this section, the dorsum linguae is made of cornified tissue and it forms a pair of dorsal cornified rods of the lingual tip (cf. Weymouth, Lasiewski & Berger, 1964). These dorsal rods become thicker and more robust when moving rostrally through cross sections (Fig. 2, XS 2–5), probably because they are the sole structural support of the rostral half of the tongue. By XS 5 there is no tissue inside the cornified semi-cylindrical grooves, and the two sides of the lingual body are completely separated (i.e., bifurcated tongue). There is almost no change between the tongue appearance and size between XS 5 and 6, which is about 3 mm corresponding to about half of the total groove length. From XS 6 to 8 there is no ostensible change in the tongue shape besides an overall reduction in size (∼25%). The rostral portion of the tongue is characterized by a reduction of the rods and a thinning in the cornified tissue comprising the grooves (Fig. 2, XS 9–10). It is worth noting that from XS 1 to 4 it is evident how the tongue fills the internal buccal spaces (when the bill is shut), leaving only a small space dorso-laterally. Such space matches the position of tongue base projections (Scharnke, 1931; XS 2 in Weymouth, Lasiewski & Berger, 1964). A reduction in the internal space outside the grooves and a tighter coupling between bill internal walls (oropharyngeal roof, or palatum, and oropharyngeal floor, or interramal region) and tongue shape is evident in the rostral portion of the feeding apparatus (Fig. 2, XS 5–10). A more in-depth description of the bill structures, such as the salivary ducts openings in the oropharyngeal floor (Fig. 2, XS 7), will be provided elsewhere.

Figure 3 Low-magnification morphology of the rostral half (grooves) of a Ruby-throated Hummingbird (Archilochus colubris) tongue.

(A) Section of the tongue embedded in resin; dorsal view oriented with the caudal end of the section at the top. (B) Corresponding cross section (light microscope) showing the semi-cylindrical configuration of the grooves. The cornified rod of the lingual tip and the outward (lateral) groove wall are labeled for reference. Unlabeled scale bars = 250 µm. (C) Histological details of the groove wall showing the stratum corneum (Sc), the strongly cornified layer (Cl). (D) Histological details of the cornified rod and the seemingly germinative layers remains.

Histology and electron microscopy

I focused on the rostral half of the tongue (e.g., Fig. 3A) to complement the work of Weymouth, Lasiewski & Berger (1964) that focused on the caudal half. At its basal region, the tongue is a cylindrical structure containing bones, muscles, vessels, blood cells, loose connective tissue, nerves, and sensory structures (e.g., taste buds), all surrounded by stratified squamous epithelium (Weymouth, Lasiewski & Berger, 1964). Moving rostrally, the tongue shape transitions into two distinct bean-shaped chambers running parallel to each other (Fig. 2, XS 1; Weymouth, Lasiewski & Berger, 1964), the paired paraglossum becomes cartilaginous and thins until it finally disappears along with the muscles, vessels, nerves, and other abovementioned structures, whereas the stratified squamous epithelium becomes thicker and a strongly cornified layer appears in between two layers of epithelium (analogous to the human nail matrix covered by the cuticle, Fig. 2, XS 2–3, 3C; Weymouth, Lasiewski & Berger, 1964). In the rostral half of the tongue all the connective tissue is absent, the bean-shaped chambers become hollow, and the remaining cornified epithelium (stratum corneum) is shaped like two extended ‘commas’ mirroring each other and forming the paired grooves or semi-cylinders at the tongue tip (Fig. 2, XS 4–10, 3B; Weymouth, Lasiewski & Berger, 1964; Ortiz-Crespo, 2003). The growing tissue seems to be abundant at the base of the grooves (cf. Fig. 2; Weymouth, Lasiewski & Berger, 1964), but to disappear in the rostral portions with few remaining cells at the interior of the cornified rod (Fig. 3D).

I found elliptical-to-circular dark corpuscles distributed evenly throughout the tongue tissue (black arrow head, Fig. 4A). The cell boundaries are continuous lines of corneo-desmosomes (e.g., black arrow, Fig. 4B). I found structures of ∼35 Å diameter that possibly are microfibrils (e.g., white arrow, Fig. 4C). Regarding the different staining methods, I found that staining with uranyl acetate and lead citrate provided the best imaging of the elliptical dark corpuscles and the most external layers of keratin, especially in the dorsal surface of the grooves (Fig. S2). However, vapor-staining with RuO4 offered the best visualization of the corneo-desmosomes necessary to study the cell architecture (Fig. S2).

Figure 4 High-magnification morphology of a cross section at the rostral half (grooves) of a Ruby-throated Hummingbird (Archilochus colubris) tongue.

(A) Transmission electron micrograph showing the difference in layer composition (more densely packed near the dorsal surface), and potential melanin (black arrow head) granules. Vapor-stained with RuO4. (B) The cellular outlines are connected corneo-desmosomes (black arrow). Stained with uranyl acetate (UA), lead citrate (LC), and RuO4 (vapors). (C) Keratinous matrix showing the microfibrils (white arrow). Stained with UA, LC, and RuO4.

In the grooved (rostral) half of the tongue, two layers of the stratum corneum can be distinguished: a thicker one underlying the ventral (convex) surface of the grooves, which I refer to as ‘cornified layer’, and a thinner one underlying the dorsal (concave) surface of the grooves (Fig. 3B). The cornified layer is made of larger cells, it is less densely packed, and it contains less granules than the layer closer to the dorsal surface (Fig. 4A). This latter layer may contain some flattened granular-cornified cells but I do not refer to it as stratum granulosum because that name is mostly applied to mammal tissues (Baumel et al., 1993). It is plausible that some of the germinative layers of this keratinized stratified squamous epithelium could be found at the basal portions of the dorsal rods (Fig. 3B), but most of it is restricted to the caudal half of the tongue (Weymouth, Lasiewski & Berger, 1964).

Probably related to the abovementioned differences in underlying tissue, I found qualitative differences between the dorsal and ventral surfaces of the tongue grooves (Fig. 5). These surfaces were cleaned in the same manner (see Methods: SEM), therefore differential accumulation of nectar or dirt residue does not appear to be a confounding factor. In addition, given that the accelerating voltage can alter the level of surface detail visualized, I kept constant 2 kV for all the comparisons. While capturing the EM images, I tried to compare corresponding points on the dorsal and ventral surface, but I did not perceive noticeable differences on the tongue surfaces depending on the relative position on the groove wall (e.g., relative distance to the cornified rod, or at the lancinated portions, Figs. 5A and 5B). At the 10-µm scale the ventral tongue groove surface (Fig. 5C) seems to have more granulated regions in comparison with the dorsal side that appears smoother (Fig. 5D). Furthermore, at the 500-nm scale the ventral surface (Fig. 5E) presented a rougher aspect than the dorsal surface (Fig. 5F).

Figure 5 Scanning electron microscopy of a Rufous Hummingbird (Selasphorus rufus) tongue.

(A) Overview of the entire tongue, although my observations focused on the rostral half (grooves). (B) Close up of a longitudinally twisted section of a tongue groove, indicating the cornified rod of the lingual tip and the lacerations of the groove wall. (C) Medium magnification (3,000×) micrograph of the ventral surface of the tongue. (D) Medium magnification (3,000×) micrograph of the dorsal surface of the tongue. (E) High magnification (50,000×) micrograph of the ventral surface of the tongue. (F) High magnification (50,000×) micrograph of the dorsal surface of the tongue. Note that when the grooves adopt their natural semi-cylindrical configuration, the ventral surface corresponds to the outer (convex) side of the groove walls, and the dorsal surface corresponds to the inner (concave) side of the groove walls.

Ex-vivo experiments

I recorded expansive filling (sensu Rico-Guevara, Fan & Rubega, 2015) in the post-mortem experiments (Fig. S1, Video S2). This observation indicates that physical (structural) rather than muscular forces are responsible for the expansion and filling of the tongue. I flattened the grooves by closing the bill tips and leaving only a small aperture to extrude the tongue through (see methods), reproducing previous observations in free-living birds (Rico-Guevara & Rubega, 2011; Rico-Guevara, Fan & Rubega, 2015), and registered that the flattened grooves expanded spontaneously upon contact with nectar in tongues of deceased specimens (Video S3). Additionally, I observed that the separation of the tips and the relaxation of the fringed regions occurred in post-mortem experiments (Video S4). Consequently, nectar trapping (sensu Rico-Guevara & Rubega, 2011) would be the first step of the fluid collecting system and is immediately followed by expansive filling. I hypothesize that the main force driving the expansive process and therefore the filling of the tongue with nectar is the elastic energy that can be stored in the cornified groove walls.

Figure 6 Elasticity-induced flow hypothesis.

(A) Dorsal photograph of a Short-tailed Woodstar (Myrmia micrura) tongue tip just after contacting the nectar surface. Given the flattened configuration of the portions of the grooves outside the nectar, there would be elastic energy stored which induces inward flow. (B) Cross section (light microscope photograph) of a hummingbird tongue in its “relaxed” configuration inside the nectar. (C) Hypothetical cross section showing the elasticity-induced flow (Ef in blue), the surface tension (γ in black), and the elastic potential energy (e in red). (D) Hypothetical cross section for a portion of the tongue not yet affected by the expansive flow. Strong nectar-wall adhesion keeps the groove in a flattened configuration, and surface tension along the groove slit prevents bubble infiltration. Elastic potential energy is larger when the bending of the wall is more pronounced; yielding a pressure differential that pumps the nectar into each groove.

I explain the hypothesis as follows: (1) The process starts when the tongue is dorso-ventrally compressed upon protrusion; when the tongue is extruded, only a thin layer of nectar remains inside the grooves. Such a thin layer acts as an adhesive (Stephan adhesion) maintaining the dorsoventrally flattened (elliptical) configuration of the grooves even after they pass the extrusion point (bill tip). The attractive forces between the nectar and the tongue (adhesion, cohesion, and surface tension) are able to resist the elastic energy stored in the grooves’ walls (cornified layers), and thus keep the grooves flattened. This stable flattened configuration is conserved during the trip of the tongue across the air space between the bill tip to the nectar pool. In the dorsal portion of the tongue, where the groove’s inside upper edge meets the rod, the free (outer) edge of the groove is prevented from rolling outward by a narrow sheet of nectar joining it to the rod. The surface tension at this exposed nectar sheet keeps the grooves “zipped up” by preventing air from entering the groove itself. Surface tension at the tip of the tongue also keeps the grooves stuck to each other, forming a unitary structure. (2) Once the tongue passes the compression point at the bill tips, there is a slight expansion in the tongue grooves (because of the cessation of compressive forces). The expansion of the grooves is arrested at the point in which the attractive forces between the tongue walls and the nectar balance out the elastic forces of the grooves’ walls. This creates an initial transient equilibrium that maintains the flattened configuration (cf. Rico-Guevara, Fan & Rubega, 2015). (3) Once the tongue tip contacts the nectar surface, the free supply of fluid eliminates the surface tension that was holding the grooves together, allowing the area of the grooves that is inside the nectar to open (cf. Rico-Guevara & Rubega, 2011). This opening of the ends of the grooves allows the nectar molecules from the nectar pool to start interacting with the nectar molecules inside the grooves (i.e., elasticity-induced flow, Fig. 6). On the dorsal surface of the length of the grooves still outside the nectar pool (more proximal to the bird’s mouth), the surface tension of the fluid sheet between the rods and the groove walls holds the grooves in the rolled, flattened position. (4) Molecules of liquid entering the tongue grooves, at the boundary where the tongue enters the nectar pool, start moving proximally through the grooves, creating a jet of fluid that fills the grooves following their expansion (cf. Rico-Guevara, Fan & Rubega, 2015). This continued destabilization of the initial transient equilibrium causes the area of the grooves outside the nectar to expand which in turn causes them to fill, creating a positive feedback that forces the grooves open along their entire length. This creates a filling front wave because the expansive process happens from the point of contact with the nectar backwards (Fig. 6). (5) The expansion stops when most of the potential elastic energy is released (and the grooves are fully reshaped into their cylindrical configuration) and when the remaining elastic energy is counteracted by the surface tension at the zipped dorsal slit (cf. Rico-Guevara & Rubega, 2011). At this point the grooves have achieved their maximum capacity, and they are completely filled with nectar.

Notes on gross tongue morphology relevant to feeding in hummingbirds

Hummingbird tongues may look as a fishing line due to their extreme slenderness, but are truly complex structures well adapted to particular tasks. Hummingbirds can extend their tongues beyond their bill tips up to about two times the bill length (e.g., Fig. 7A), given that most hummingbird tongues are only slightly longer than their bills (Fig. 2, Rico-Guevara, 2014), the tongue base can be extended pass the bill tip (transition visible in Fig. 7A). This remarkable lingual protraction can be achieved by the rostral displacement of the elongated hyoid apparatus (e.g., Video S5 ), and hummingbirds can protrude their tongues with their bills closed because of the presence of an elastic envelope between the larynx and the tongue base (e.g., Fig. 7B), which allows lingual protraction without dragging the trachea inside the bill. Lingual protrusion serves to increase the range of the tongue tips, and also to reach the bill tips with the tongue base, which is important for the intraoral transport of the food. At the tongue base, hummingbirds present two caudal-facing flaps without conical papillae (e.g., Fig. 7C), which may aid during intraoral transport. I did not find papillae neither through macroscopical observations of the entire tongue nor through microscopical observations at the rostral regions.

Figure 7 Gross morphology of hummingbird tongues.

(A) Photograph of a Fawn-breasted Brilliant (Heliodoxa rubinoides) stretching its tongue apparatus (courtesy of Jim DeWitt –Frozen Feather Images). (B) Dissecting microscope photograph of the throat region in a dissected specimen of a White-necked Jacobin (Florisuga mellivora) showing the accordion-like structure or tuba elastica in its retrieved position. The tuba elastica can contain the basihyal and ceratobranchial bones allowing them to move independently from the surrounding tissue and permitting the extreme protraction of the tongue. (C) Macro photograph of the bill and tongue-base of a White-tipped Sicklebill (Eutoxeres aquila). Note the alae linguae at the base of the tongue (black arrow), which are enlarged in comparison to other hummingbirds.

Discussion

Gross morphology of hummingbird tongues

Hummingbird tongues entirely lack papillae, a rare condition in vertebrate tongues (Schwenk, 2000; Iwasaki, 2002) and even among birds (review in Erdoğan & Iwasaki, 2014). Avian lingual papillae are involved in manipulation of solid food (e.g., prey apprehension, holding, cutting, filtering, shelling, Iwasaki, Asami & Chiba, 1997; Kobayashi et al., 1998; Jackowiak et al., 2010; Jackowiak et al., 2011; Guimarães et al., 2014; Skieresz-Szewczyk & Jackowiak, 2014) and caudal intraoral transport of solid items (review in Parchami, Dehkordi & Bahadoran, 2010). Hummingbirds have remarkable feeding modes; first, about half of their diet (cf. Stiles, 1995) is composed of floral nectar that is collected inside the tongue grooves; this process does not involve adhesion of the liquid to intra-papillar spaces, as in the case of bats (Birt, Hall & Smith, 1997; Harper, Swartz & Brainerd, 2013) or lorikeets (Homberger, 1980, p. 41). Second, the other half of their diet (cf. Stiles, 1995) consists of arthropods, which most hummingbirds capture by flycatching (Stiles, 1995; Rico-Guevara, 2008). Yanega & Rubega (2004) showed that the flycatching mechanism in hummingbirds involves an expansion of the gape (see also Smith, Yanega & Ruina, 2011) and most of the aerial prey are captured at the base rather than at the tip of the bill; therefore, little or no intraoral lingual transport is necessary. Other hummingbirds, especially from the subfamily Phaethornithinae (‘hermits’), consume mostly substrate-captured prey (e.g., spiders, Stiles, 1995). This is also the case of reproductive females of many species across the family, which have higher protein requirements (Rico-Guevara, 2008; Hardesty, 2009). In the process of consuming substrate prey or prey that are generally captured near the bill tip, hummingbirds, as other birds, can use inertial transport (cf. Mobbs, 1979; catch and throw, Zweers, Berge & Berkhoudt, 1997; or cranioinertial feeding, Tomlinson, 2000; Gussekloo & Bout, 2005; also called ballistic transport, Baussart et al., 2009; Baussart & Bels, 2011; Harte et al., 2012) while flying, or lingual transport (Yanega, 2007). Hummingbirds have evolved the ability to protract their tongues past the bill tips to feed on nectar, but the purpose of the extreme protrusion that they can achieve (e.g., Fig. 7A) is still a mystery. Thus, hummingbirds can reach the rostral portions of their bills with the tongue base (to perform lingual transport for instance), without dragging their tracheae rostrally, because of the development of an accordion-like tube (tuba elastica, Zusi, 2013) between the epiglottis and the tongue base which can contain a large part of the hyobranchial apparatus during tongue protrusion (cf. Weymouth, Lasiewski & Berger, 1964; Fig. 7B). This tuba elastica appears to be a modification of the fibrous attachment between the rostral process of the cricoid cartilage and the rostral process of the basihyale (Soley, Tivane & Crole, 2015). Hummingbirds’ lack of lingual papillae and protrusion abilities may be explained by their arthropod hunting and consumption strategies, as well as their liquid food collecting method: grooves with smooth surfaces are easier to extrude nectar from, and protrusible tongues not only to reach but also to transport food intraorally.

Besides lacking papillae, hummingbird tongues are also unique because of their alae linguae (cf. Weymouth, Lasiewski & Berger, 1964; Homberger, 2017), which are flattened projections at the base of the tongue (Fig. 7C). These two flaps are located and oriented at the same place and in the same general direction as the papillary crest in other birds. Nevertheless, these structures do not present caudally directed conical papillae, as is usual in avian tongues (e.g., Erdoğan & Alan, 2012; Erdoǧan, Pèrez & Alan, 2012). In comparison to the width of the tongue, these flaps are greatly elongated laterally in Sicklebill hummingbirds (Eutoxeres, Fig. 7C), which have strongly decurved bills. These flaps are thin and flexible at touch, as well as positioned dorso-laterally forming a V-shaped structure. These flaps in hummingbirds have no parallel among nectar-feeding birds (Lucas, 1894; Scharnke, 1932; Scharnke, 1933; Rand, 1961; Rand, 1967; Bock, 1972; Morioka, 1992; Pratt, 1992; Downs, 2004; Chang et al., 2013), or birds in general (e.g., Erdoğan & Alan, 2012; Erdoğan, Sağsöz & Akbalik, 2012; Erdoǧan, Pèrez & Alan, 2012; Erdoğan & Iwasaki, 2014; Erdoğan & Pérez, 2015). I hypothesize that the alae linguae could aid to move the nectar backwards during its intraoral transport (Rico-Guevara, 2014) and to drag proximally arthropod prey that are caught at different places along the bill length (cf. Yanega, 2007). In terms of general shape, hummingbird tongues are not triangular and dorsoventrally flattened as in most birds (review in Erdoğan & Pérez, 2015), instead, as it is the case in other nectarivorous birds, these tongues are cylindrically shaped (e.g., Bock, 1972; Downs, 2004; Chang et al., 2013). Lastly, I found that hummingbird tongues near the tip also lacked taste buds and salivary glands (found in other birds, review in Erdoğan, Sağsöz & Akbalik, 2012), in agreement with previous work by Weymouth, Lasiewski & Berger (1964).

Ultrastructural characteristics of hummingbird tongues

The rostral portions of the hummingbird tongue, the ones that collect the food, are mostly transparent and their tissues are extremely thin (Figs. 2, 8A and 8C), a rare condition in vertebrates. The species studied with TEM had transparent tongues and also presented few and small dark corpuscles (Fig. 4A), which possibly are melanin granules (e.g., Dummett & Barens, 1974). I expect that species with darker tongues (tongue color varies across the family, (Rico-Guevara, 2014)) will have more and/or larger dark corpuscles of the kind reported here. The ∼35 Å diameter structures that I found in the tissue (Fig. 4C) are likely to be microfibrils; the ventral layers of cornified tissue are more similar to those found in feathers (β-keratin) than to that of tissues with α-keratin (cf. Filshie & Rogers, 1962). Specifically, the diameter of the putative microfibrils is within the range of other β-keratin tissue microarchitectures (Parakkal & Alexander, 1972, p. 33), and almost a third of the diameter of α-keratin microfibrils (Filshie & Rogers, 1962; Johnson & Sikorski, 1965). In most avian tongues the stratum corneum at the ventral surface comprises less than 10% of the lingual tissue in a cross section (Erdoğan, Sağsöz & Akbalik, 2012; Erdoğan & Iwasaki, 2014). Different from most birds, the cornified ventral layer in hummingbirds accounts for between 50%, near the cornified rod and near the groove base, and 100%, at the edge of the groove wall and at the tongue tip, of the tissue in cross sections (Figs. 2, 3A, 8B and 8D, S2). I suggest that most of the germinative layers of this keratinized stratified squamous epithelium (including the layers of dead cells, the stratum corneum) disappear before reaching the most rostral portions of the hummingbird tongue; similar to what would be expected in cross sections of human nail overhangs. Therefore, the caudal half of the hummingbird tongues is made of dead cornified tissue that is shaped by the interaction with the bill, and it is constantly replaced from the rostral half. A thick (cornified) layer of β-keratin can increase mechanical resistance on a surface that is compressed and scraped by the serrated edges of the bill tip ∼14 times a second (Ewald & Williams, 1982) and literally tens of thousands of times a day (Rico-Guevara, 2014). Future experiments to test the hypothetical high percentage (50–100%) of β-keratin in the hummingbird tongue grooves could use in situ hybridization, immunolabeling for β-keratins (e.g., in Alibardi et al., 2009) or selective biodegradation of β-keratin (e.g., Lingham-Soliar, Bonser & Wesley-Smith, 2010; Lingham-Soliar & Murugan, 2013).

Figure 8 Tongue groove morphology at the most distal portions (near the tip) in a Ruby-throated Hummingbird (Archilochus colubris).

(A) Photograph showing the tongue protrusion, its bifurcation, and the relaxed morphology of the grooves inside the nectar (courtesy of Don Carroll). (B) Cross section (light microscope) showing the reduction in cornified rod diameter and the thinning in the stratum corneum composing the grooves (which at this point is composed only of the cornified layer). (C) Close up to the tongue tip showing the membranous appearance of the grooves and the presence of diagonal cuts in the tissue (lancinated groove walls). (D) Electron micrograph showing the structure of the cornified layer, note the reduction in the number of cell layers and the absence of delineated boundaries in the dorsal surface (on top).

I found differences between the layers of tissue underlying the dorsal and ventral surfaces of the tongue grooves (Fig. 3B). These differences may be explained by the organization of the tissues (Fig. 4A), but they may also be influenced by differential composition and organization between proteins (fibrous vs. matrix components) and/or the presence of β-keratin (reviewed by Alibardi et al., 2009), which has been found in the rostral ventral epithelium of other avian tongues (review in Carver, Knapp & Sawyer, 1990). On the ventral surface of the tongue grooves I found thick stratum corneum (cf. Fig. 4 in Kadhim et al., 2013; Figs. 5 and 6 in Jackowiak et al., 2015), but without the underlying lamina propria characteristic of heavily cornified areas in bird tongues (Farner, 1960; Kadhim et al., 2013). This stratum corneum in the tongue surface is common in birds (Farner, 1960; Erdoğan, Sağsöz & Akbalik, 2012; Erdoğan & Iwasaki, 2014), however, as opposed to hummingbirds, in several bird species the stratum corneum is better developed on the dorsal lingual surface (Iwasaki, 2002; Erdoğan, Sağsöz & Akbalik, 2012). I found more sloughing cell layers in the histology and TEM preparations in the dorsal compared to the ventral surface, which indicates that the ventral surface is underlain by harder keratin (cf. Lucas & Stettenheim, 1972). Interestingly, my results are consistent with the idea that dorsal and ventral surfaces of hummingbird tongues have different rugosities (Figs. 5 and 8D). To conclude that there are significant differences between dorsal and ventral surfaces of the hummingbird tongue, it would be necessary to quantify differences in roughness; the best way to do this is by using Atomic Force Microscopy (e.g., Ghosh et al., 2013). Alternative techniques (e.g., Nanda, Sarangi & Sahu, 1998; Fujii, 2011; Kremer et al., 2015) include the use of optical interferometry (e.g., white light scanner), and 3-D reconstructions of tilted SEM micrographs (stereomicroscopy). Differential rugosity between tongue surfaces would have direct implications for their hydrophobicity, i.e., increased roughness may significantly increase contact angle of a water droplet and decrease contact angle hysteresis, which would augment its hydrophobicity (e.g., Michael & Bhushan, 2007). Therefore, the dorsal tongue groove surface, which is less rugose, may be more hydrophilic than the ventral grove surface, and potentially facilitating the fluid trapping process described by Rico-Guevara & Rubega (2011).

Microanatomy of the hummingbird feeding apparatus

Hummingbird tongues, as well as most avian tongues, correspond to the shape of the interramal region (oropharyngeal cavity floor), although commonly not to its size (e.g., Abou-Zaid & Al-Jalaud, 2010; Tivane et al., 2011; review in Abumandour, 2014). Nevertheless, it is worth noting that avian tongues are not larger than the oropharyngeal cavity (as it is the case in some nectarivorous bats, Muchhala, 2006), instead, to reach farther away from the tip of their bills, the mobile bones of the hyoid apparatus in some avian taxa appear greatly elongated, allowing for tongue protrusion (e.g., Video S5). In hummingbirds, the tongue grooves fit perfectly in the rostral portion of the oropharyngeal cavity and match both lower and upper bill internal walls (Fig. 2), which is of vital importance for the efficient offloading of nectar (cf. Rico-Guevara & Rubega, 2011) and intraoral transport (Rico-Guevara, 2014). My study presents the first high-resolution (5-µm voxels) CT scan of a vertebrate tongue satisfactorily stained to highlight soft tissue. A study on flamingos presented detailed CT scans of the head (including the tongue) stained with a novel injection technique (Holliday et al., 2006), but it focused on vascular anatomy at lower resolution than in the present study. Within the last five years other studies have used a variety of techniques to enhance visualization of soft tissue in vertebrates (reviews in Gignac & Kley, 2014; Lautenschlager, Bright & Rayfield, 2014; Gignac et al., 2016), but they have not been focused on tongues or worked at the micro scale of the present study. This three-dimensional modeling of hummingbird tongues allows for the clarification of some misconceptions; for instance, it has been suggested that the mathematical model derived for capillary filling provides a rationale for the shape of hummingbird tongues (Kim et al., 2012). Specifically, that the semi-cylindrical shape of the grooves (cylinders with a dorsal slit) can be explained by an optimal opening angle of a cross section, which matches a peak of energy intake rates (Fig. 4 in Kim et al., 2012). I prefer a more parsimonious explanation: starting with a dorso-ventrally flattened tongue as an ancestral condition (cf. Emura, Okumura & Chen, 2010; Shah & Aziz, 2014), evolution would maximize the nectar-holding capacity by selecting for a cylindrical structure. In the same way in which a sphere is the shape with the lowest surface area to volume ratio, for an elongated structure (like a tongue), a cylindrical configuration achieves the greatest capacity for a given amount of tissue (in this case, the groove walls). It is worth noting that the tongue tip whilst outside the nectar ends in a conical shape (Fig. 1 in Rico-Guevara, Fan & Rubega, 2015), due to a shortening of the cross-sectional length of the groove wall (Figs. 2 and 3), which helps to trap and retain the nectar at high licking rates (Rico-Guevara & Rubega, 2011). Rostrally, the groove wall membranes exhibit diagonal to perpendicular cuts in the tissue starting from their lateral edges (Fig. 8C), forming lancinated walls in the distal portions of the grooves (Lucas, 1891; also called lamellae, Rico-Guevara & Rubega, 2011). Such cuts may originate by wear during the extruding action of the serrated bill tips on the rostral tongue portions (Lucas, 1891; Rico-Guevara, 2014), and may facilitate the bending of the tongue tip and trapping of fluid drops while mopping the inside of nectar chambers. Wearing at the tongue tip seems to counteract the continuous elongation of the tongue by the growing tissue at the base of the grooves (cf. Fig. 2), and unpublished descriptions of hummingbirds with ‘dislocated’ tongues (feeding from artificial feeders with the tongue always hanging to one side from the bill base) report that their tongues are unusually long and/or they become longer with time.

Additionally, microCT data could inform the mathematical models necessary to make predictions about feeding efficiency across the varying morphology of hummingbird species. For instance, by calculating the total and partial groove capacities depending on immersion lengths (conditioned by the nectar pools on the flowers they visit) the expected amount of liquid extracted can be obtained, and then compared to performance measurements in the wild. Further calculations of the intraoral flow on nectar (based on the bill-tongue internal coupling) taking into account a range of liquid properties that vary in nature (e.g., composition, viscosity, temperature, etc.) will provide information on the limiting step of the fluid collection and transport system. Such an approach would generate falsifiable quantitative predictions about the action of the feeding apparatus, and the volumes of nectar that can be collected and the speed at which they can be transported, for nectars of different concentrations and at different temperatures (hummingbirds feed from flowers at elevations as high as 5,000 m, Carpenter, 1976). Results from this proposed approach will shed new light on the long-standing debate about the reason of the mismatch between hummingbird nectar concentration preferences (Hainsworth & Wolf, 1976; Roberts, 1996; Morgan et al., 2016) and the concentration of the nectar of the flowers they pollinate (review in Nicolson, Nepi & Pacini, 2007). The predictions from these mathematical models available only with the MicroCT reconstruction data, could be tested with additional experiments under controlled conditions using post mortem tongues (building on the ex-vivo experiments presented here), and by measuring nectar extraction rates (fluid volume uptake (µl/s)) in free-living nectarivores living under extreme environmental conditions.

Biophysics of nectar collection

The post-mortem observations (e.g., Videos S3, S4) are consistent with the idea that expansive filling and nectar trapping are processes that do not incur any extra energy than that necessary to squeeze the nectar out of the tongue and inside the bill, making this elastic micropump a highly efficient device (Rico-Guevara, 2014). This is because when the tongue tips enter the surface of the nectar pool, the attractive forces (adhesion and cohesion) holding the groove walls flattened get weaker because more molecules of fluid are available to fill the internal groove space. This creates an imbalance, with elastic forces dominating, which results in reshaping of the groove walls away from the flattened configuration at the tongue tips. Molecules of nectar are pulled inside the grooves through the release of the elastic energy initially stored on the flattening of the groove walls (Fig. 6). Because the grooves are sealed on top (by surface tension in the zipped dorsal slit), the release of the elastic energy (reshaping of the grooves) pulls more and more nectar molecules inside the grooves until they reach a stable cylindrical configuration, from the tips to the base of the grooves. The net result of this process is that the portions of the tongue that remain outside the liquid expand and are filled quickly with nectar, thereby improving fluid collection efficiency. Thus, the tongue filling is achieved through the transition from a high potential energy state (flattened grooves) to a low potential energy state (filled grooves). In summary, the elastic properties of the cornified layer make the elasticity-induced flow hypothesis plausible. This is ecologically relevant because when the bill tip is almost in contact with the nectar surface (most likely scenario in the wild given hummingbird flowers’ internal morphology), the process described above is sufficient to fully load the fringed distal portion of the tongue. Nevertheless, when the bill tip is not in contact with the surface of the nectar (e.g., hummingbirds visiting flowers with corollas longer than their bills), but instead there is a space between the bill tip and the nectar pool, the portion of the tongue that remains outside the liquid would be filled with fluid by the interaction of the aforementioned physical forces in a process I hypothesize as follows: As the tongue is protruded, the grooves are dorso-ventrally flattened by the bill tips, and once the tongue tip contacts the nectar surface the fluid starts to penetrate the flattened grooves (because of cohesion of water molecules in the nectar pool and water molecules in the nectar remaining trapped inside the tongue). When the grooves expand, their walls start releasing the potential energy stored by the bending (flattening by the bill tips). At this point, the excess Laplace pressure due to the nectar flowing inside the grooves plus the releasing of the potential energy whilst the grooves’ walls are recovering their semi-cylindrical shape, create a positive feedback between the groove’s internal space expansion and the nectar flow. The net result of this process is that the portion of the tongue that remains outside the nectar is also loaded with nectar (Fig. 6). Additionally, if there are empty portions of the tongue still inside the beak before passing through the squeezing zone at the bill tip, the nectar filling the grooves will reach them via cohesive and adhesive forces to all the internal surfaces; thereby allowing complete loading of the grooves. Alternatively, the complete filling of the tongue may be achieved by the bill-tongue interaction, involving mechanisms like suction, surface tension transport, hydrostatic pressure motion, etc. However, this would be dependent on, and pertains to, the intra-oral transport of the nectar, which remains understudied.

Conclusions

A variety of anatomical structures allow hummingbirds to protrude their tongues and drag food backwards. Hummingbird tongue shape matches the shape of the internal bill walls, which is important to understand and model the squeezing of the tongue and movement of the nectar to the throat. The rostral portions of the tongue are mostly made of a cornified layer (β-keratin) that is replaced from the tongue basal portions, and worn at the tip by the interaction with the bill tips upon nectar extrusion. Interestingly, if the dorsal and ventral surfaces have different rugosities that may have direct implications to their hydrophobicity, i.e., increased roughness may significantly increase contact angle (of a water droplet) and decrease contact angle hysteresis (e.g., Michael & Bhushan, 2007). Therefore, at the grooves, the inner tongue surface may be more hydrophilic than the outer surface, potentially helping the fluid trapping process (Rico-Guevara & Rubega, 2011) and maintaining the surface tension zip at the dorsal slit along the grooves (Figs. 6C–6D).

Hummingbird tongues are thinner than other bird tongues (references above), the walls of the grooves are between ∼10 and 30 µm thick, which makes them highly pliable. In addition, the tissue architecture of the cornified layer resembling a brick-wall configuration, along with its keratinous composition, grants non-stretchable properties to the grooves. Hence, hummingbird tongues are easily squeezed to unload the nectar inside the bill (Rico-Guevara & Rubega, 2011), yielding to storage of elastic potential energy in the flattened tips, which is then released when the tongue is reinserted in the nectar (Rico-Guevara, 2014), thereby improving liquid uptake efficiency. The proper functioning of hummingbird tongue grooves as dynamic structures depends on the balance between pliability and elasticity; in particular, the latter has to be strong enough to help the pumping process to extract nectar but weak enough to keep the grooves flattened until they contact the nectar surface (Rico-Guevara, Fan & Rubega, 2015). Several scaling models and applications have been developed on the basis of recent discoveries of biological phenomena and underlying physical explanations (see Vogel, 2011), which opens the way for deeper studies of the influence of the surface characteristics (e.g., differential hydrophilicity) and the tissue composition of the grooves on the elastic properties of hummingbird tongues.

The present work raises anew the question: how do hummingbirds feed? Much work remains before the whole nectar feeding process in hummingbirds and other nectarivores can be fully explained. Achieving a fuller understanding of the mechanics of the nectar-feeding process may help eliminate the disparity between the theoretical predictions of how birds should act and empirical observations of what they actually do. A detailed three-dimensional morphological description that allows for detailed mathematical modeling will aid in understanding different aspects of their food collection efficiency limits and deviations of predicted vs. realized performance, which are the building blocks of foraging and coevolution principles (review in Pyke, 2016). Since the inferences presented in this paper apply only to the species studied, future work should focus on corroborate or disprove the trends presented here applying equivalent methods on a wider range of taxa. Detailed accounts on the gross morpho-functional diversity of the feeding apparatus of hummingbirds have been accomplished in the past (e.g., Yanega, 2007; Rico-Guevara, 2014), but detailed comparative and phylogenetically corrected studies including modern visualization techniques are warranted (e.g., CT scans, Ekdale, 2006; 3D white-light scans, Cooney et al., 2017). This paper sets the bases for morpho-functional comparisons between hummingbirds and other nectar feeding organisms, as an example of convergent and alternative ways to maximize food collection efficiency in nature.

Supplemental Information

Supplemental Information 1 Supplementary Information

Legends in the file

Click here for additional data file.

Video S1 MicroCT rendering (rostro-cranial coronal cross sectioning) of the bill and tongue of an Anna’s Hummingbird, overlaying TEM micrographs

This virtual model of the internal three-dimensional architecture help us to understand the fit between bill and tongue. Additionally, merging microCT, light, and electron microscopy allows us to understand the key morphological features for the tongue functioning linking them across spatial scales. Known objects are placed at the same scale at the different zoom levels in order to contextualize the structures shown.

Click here for additional data file.

Video S2 Control of the setup to film post-mortem tongue fluid interactions

Under a dissecting microscope (cf. Fig S2), this control system emulates reciprocating tongue movements and allows for fine control of the distance to the nectar surface using micromanipulators. I coupled a high-speed camera to capture the details of tongue fluid interactions (e.g., Video S3).

Click here for additional data file.

Video S3 High-speed recording of post-mortem tongue expansive filling

A slow motion (165 times slower than real time) video of the lateral view of a dissected Ruby-throated Hummingbird focusing on the bill tip—tongue—nectar interaction. The tongue protraction is controlled by micromanipulators (Video S2). Footage obtained under a dissecting microscope (cf. Fig S2) with a flat surface mirror to achieve the side view.

Click here for additional data file.

Video S4 Recording of post-mortem tongue fluid trapping

A real-time video of the dorsal view of a deceased Ruby-throated Hummingbird focusing on the bill tip—tongue—nectar interaction. Footage obtained under a dissecting microscope immersing the tongue in a nectar reservoir to appreciate the instantaneous change in shape when the tongue transitions form air (out of focus) to nectar (in focus) and vice versa.

Click here for additional data file.

Video S5 Three-dimensional digital rendering of a microCT scan of the skull of a Ruby-throated Hummingbird

This spinning reconstruction makes it possible to follow and visualize the elongated epibranchials surrounding the skull. In this case, the tongue is retracted inside the bill.

Click here for additional data file.

I thank Margaret Rubega, Diego Sustaita, and Kurt Schwenk for thorough discussions; Stephen Daniels and Marie Cantino for their help with electron microscopy and specimen staining for microCT; Kristiina Hurme for style corrections; the National Science Foundation funded course: Basics of CT data acquisition, visualization, and analysis, at The University of Texas High-Resolution X-ray CT Facility for training; Dominique Homberger, Robert Colwell, Tai-Hsi Fan, and Carl Schlichting for their priceless comments on earlier versions of the manuscript; and the Miller Institute at UC Berkeley.

Additional Information and Declarations

Competing Interests

Author Contributions

Animal Ethics

Data Availability

The author declares there are no competing interests.

Alejandro Rico-Guevara conceived and designed the experiments, performed the experiments, analyzed the data, contributed reagents/materials/analysis tools, wrote the paper, prepared figures and/or tables, reviewed drafts of the paper.

The following information was supplied relating to ethical approvals (i.e., approving body and any reference numbers):

Activities were reviewed and authorized by the Institutional Animal Care and Use Committee at the University of Connecticut.

The following information was supplied regarding data availability:

The raw data is included in the figures and in the Supplemental Files.

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
