# Peer review of "Relating form to function in the hummingbird feeding apparatus"

_PeerJ, doi:10.7717/peerj.3449_

## Round 0.1 · original submission · Minor Revisions

· Academic Editor

Minor Revisions

First of all I would like to apologize for the delay in processing this ms. I have two reviews of your manuscript that find it quite interesting - both have provided additional PDFs. I think, however, that it needs more work to be suitable for publication. I am concerned about the lack of consideration of several recent papers on bird beak and skull morphological evolution. Please address the problem of the phylogenetic relationships of the six analyzed species.

·

Basic reporting

I found only one reference not cited in the text.

All else ok.

Experimental design

All ok.

Validity of the findings

Detailed descriptions reasonably clear.
Results fine.
Conclusions well stated.

Additional comments

Good job, very thorough and well researched (good literature review).
Most of my edits were grammatical or cosmetic for easier reading.
The descriptions were detailed and thorough.
Functional interpretations seem well supported.

·

Basic reporting

The paper presents a broad structural and functional study of the unusual tongue structure of hummingbirds, focusing on ten specimens of six species. A range of imaging and ex-vivo functional approaches are employed to understand the tongue at different levels of resolution, from the macroscale to submicron detail.

The first paragraph of the introduction decribes how bill shape is tightly correlated to diet, and cites Rubega 2000. There have been recent studies - Bright et al. 2016 that dispute this tight link between bird skull shape and function, and describe how allometry and phylogeny also play key roles. Furthermore, the recent 2017 Cooney, Bright et al. paper in Nature highlights how hummingbirds show rapid rates of diversification againts background rates in birds. The introduction needs to demonstrate a broader recent knowledge of advances in bird beak and skull morphological evolution.

It is assumed throughout the text that the six species demonstrate similar structural and functional characteristics, but surely there must be some interspecific variation? How did you control for this during your analysis? And how are these six species related? Do they share a recent common ancestor or not?

Experimental design

The methods are clearly described and repeatable. They are appropriate for the various aims of the study and to address the overall aim of the research. The rationale for the study is stated, especially the focus on the tip of the tongue and how its structural properties contribute to nectar uptake.

Validity of the findings

The findings are extremely interesting, novel and clearly presented. Inferences of feeding function are drawn from the ex-vivo work. Inference is clearly stated and the conclusions are not overblown but pave the way for future research.

Additional comments

A very interesting and readable paper. Nothing further to add.

---

## Round 0.2 · Minor Revisions

· Academic Editor

Minor Revisions

Although I am inclined to accept your work some minor issues require of your further attention. Please note that in your results section many paragraphs of discussion are interspersed. For example lines 529 to 540 among many others. Presenting your results free of discussion paragraphs would contribute a lot to reading fluency.

---

## Round 0.3 · accepted · Accept

· Academic Editor

Accept

Thank you for considering all the suggestions we made. This is truly a nice paper.